# Structural Characterization of a Neutral Polysaccharide from *Cucurbia moschata* and Its Uptake Behaviors in Caco-2 Cells

**DOI:** 10.3390/foods10102357

**Published:** 2021-10-03

**Authors:** Fei Li, Solju Pak, Jing Zhao, Yunlu Wei, Yuyu Zhang, Quanhong Li

**Affiliations:** 1College of Food Science and Nutritional Engineering, China Agricultural University, Beijing 100083, China; pdlifeiyyds@cau.edu.cn (F.L.); lb20193060027@cau.edu.cn (S.P.); zhaojing@cau.edu.cn (J.Z.); weiyunlu@cau.edu.cn (Y.W.); 2Beijing Key Laboratory of Flavor Chemistry, Beijing Technology and Business University (BTBU), Beijing 100048, China; zhangyuyu@btbu.edu.cn

**Keywords:** *Cucurbia moschata*, polysaccharide, structural characterization, uptake characteristic

## Abstract

A neutral pumpkin polysaccharide (NPPc) was extracted from *Cucurbia moschata* and its structural characterization is performed. Moreover, uptake behaviors of an NPPC were investigated at the cellular level. The results showed that NPPc, an average molecular weight (Mw) of 9.023 kDa, was linear (1→4)-α-D-Glc*p* residues in the backbone, which branched point at *O*-6 position of (1→4,6)-α-D-Glc*p*. The side chain contained (1→6)-α-D-Glc*p* and terminal glucose. The cellular uptake kinetics results showed that the uptake of fluorescent-labeled NPPc was in time- and dose-dependent manners in Caco-2 cells. For subcellular localization of NPPc, it was accumulated in endoplasmic reticulum and mitochondrion. This study illustrates the characteristics on the uptake of NPPc and provides a rational basis for the exploration of polysaccharides absorption in intestinal epithelium.

## 1. Introduction

Pumpkin, family Cucurbitaceace, is widely cultivated in the world and plays an important role in Traditional Chinese Medicine [1]. It is rich in various bioactive ingredients, including polysaccharides, proteins, vitamins, flavonoids and so on [2]. Many previous reports demonstrated that pumpkin polysaccharides play important roles in the hypoglycemic, anticarcinogenic, anti-inflammatory, and antioxidant bioactivities [3,4,5,6]. Recently, polysaccharides obtained from fungus, animals, and plants have caused more and more attention because of their functional potential, which is closely related with the corresponding structures [7]. Notably, monosaccharide compositions, types of glycoside bonds, spatial conformation, the molecular weight, and other structural characteristics of polysaccharides are connected with multiple bioactivities [8]. Therefore, structural elucidation of pumpkin polysaccharides is of great significance. So far, researches on pumpkin polysaccharides mainly focus on acid polysaccharides [9,10,11,12,13,14]. For studies of neutral polysaccharides, Chen et al. [5] obtained a neutral pumpkin polysaccharide extracted by aqueous two-phase system, which was constituted by (1→3)-linked-Glc*p* as backbone. Unfortunately, pumpkin species was not mentioned in their study. In our previous studies, neutral polysaccharides by hot-water extraction from *Cucurbita pepo* lady godiva have been identified to mainly consist of α-(1→6)-galactose, α-(1→4,6)-glucose, α-(1→3)-glucose, and terminal glucose [15]. Later research indicated that a neutral *C**ucurbita maxima* polysaccharide fraction was composed of (1→4)-galactose [16]. However, neutral polysaccharides’ structure of *Cucurbita moschata* extracted by hot-water has not yet been elucidated. Up to now, precise structural features of neutral polysaccharides from *Cucurbita mostacha* are still unknown. To further extend the understanding of Cucurbita genus and compare the structural features of carbohydrate polymers from pumpkin in Cucurbita genus, exploration on the structure characterization aspect needs to be, further, conducted.

It was well known that the pharmacological activities were effective through gastrointestinal administration of pumpkin polysaccharides, nevertheless, relevant mechanisms were still no clear. As for most oral agents, absorption into the blood circulation system through gastrointestinal tract is the precondition for nutrients bioavailability and comprehending bioactivities mechanisms [17]. To reveal bioactivities mechanisms of oral administration polysaccharides, researchers have inclined to use a variety of tissue cell culture models (e.g., Caco-2 monolayer cells model) to simulate intestinal epithelial cells [18]. As hard as it is, limited knowledge was found about its study of intestinal epithelium absorption because of a lacking of sensitive detection methods and complicated polysaccharide structures. Caco-2 cells are derived from human rectal cancer and colon cancer and its structure and function are similar to those of human intestinal epithelial cells. Under particular culture environments, it can perform biochemical and morphological differentiation in vitro, and produce microvilli structure and small intestinal brush border epithelial related enzymes [19]. Thus, the Caco-2 cells have begun to be used as cell culture models in vitro transport characteristics and bioavailability of nutrients or drug molecules by evaluating some important parameters. Because of its advantages of good repeatability, low cost, and high accuracy, in vitro research is currently widely used. Wang et al. [20], Zhang et al. [21], and Xiang et al. [18] studied the transport characteristics of angelica polysaccharides (cASP), fucoidan sulfate, and Se-enriched *Grifola frondosa* polysaccharides (Se-GFP-22), respectively, in Caco-2 cells in vitro. The results exhibited that the transport situation into the intestinal cells are different for different structural polysaccharides and physicochemical properties. Uptake is the first step in the intestinal transport process. Thus, we investigated whether pumpkin polysaccharide could be uptake by Caco-2 cells, and this study provides a favorable basis for whether NPPc can be absorbed by small intestinal epithelial cells into the blood circulation.

In this study, a neutral polysaccharide fraction was obtained from *Cucurbita mostacha*, and its structure was characterized using multiple means and methods, such as Fourier transform infrared spectroscopy (FT-IR), high performance gel permeation chromatography (HPGPC), determination of monosaccharide composition, methylation analysis, and nuclear magnetic resonance (NMR) spectroscopy. Furthermore, NPPc was conjugated with fluorescein isothiocyanate (FITC) to investigate the uptake of pumpkin polysaccharides by Caco-2 cells, flow cytometry, and laser confocal microscope. This article might provide some information on understanding of polysaccharides from the *Cucurbita mostacha* and facilitate their applications in the fields of nutraceutical foods and medicine.

## 2. Materials and Methods

### 2.1. Plant Materials and Chemicals

Fresh pumpkins (*Cucurbita moschata*) were purchased from a local commercial mass (Beijing, China). Pumpkins were washed with a laboratory mill and the peel and seeds were discarded. Dulbecco’s modified Eagle’s medium (DEME), penicillin, streptomycin were obtained from Gibco (Gibco Life Technologies, Grand Island, New York, NY, USA). Tyramine, sodium cyanoborohydride (NaBH_3_CN), and FITC were collected from Aladdin (Aladdin Biochemical Technology Co., Shanghai, China). For this investigation, 24, 12, and 6 well plates were purchased from Corning (Corning Incorporated, New York, NY, USA). ER-Tracker Red and Mito Tracker Red were obtained from Beyotime (Beyotime Institute of Biotechnology, Shanghai, China). DEAE cellulose-52 was acquired from Solarbio (Beijing Solarbio Science & Technology Co., Ltd., Beijing, China). Other chemicals and solvents were analytical or chromatographic grade.

### 2.2. Extraction and Purification of NPPc

Samples were extracted in distilled water as extracting agent with a solid–liquid ratio of 1/4 (*w*/*v*). The mixture was heated at 90 °C for 6 h with constant stirring. After centrifugation, the supernatant was collected, and concentrated to 200–300 mL by evaporation. Subsequently, proteins in the supernatant were removed by the Sevag reagent method. Ethanol was added into the solution to reach the final concentration of 85 vol%. Then, the solution was precipitated at 4 °C for 24 h, followed by centrifugation at 10,000× *g* for 15 min to collect the precipitate. Precipitate was redissolved into deionized water, dialyzed, and, finally, lyophilized. Then, final precipitated polysaccharide was collected by the method of a stepwise fractionated precipitation with absolute ethanol. Briefly, absolute ethanol was poured to the polysaccharide solution with vigorous stirring to final concentration of ethanol reached 65%. Then, supernatant solution was collected by centrifuging at 10,000× *g* for 15 min. After centrifugation, the supernatants were concentrated to a quarter of the original volume by evaporation, and lyophilized (named as NPPc).

### 2.3. Homogeneity and Molecular Weight of NPPc

The homogeneity and the weight-average molecular weight (Mw), number-average molecular weight (Mn), and molecular weight distribution (Mw/Mn) of NPPc were analyzed through an HPLC system (Shimadzu LC-10A, Shimadzu Co., Kyoto, Japan) using a BRT105-104-102 (8 mm × 300 mm, BoRui Saccharide Biotech Co. Ltd., YangZhou, China) column and equipped with a refractive index (RI) detector. Sample (20 μL) solution (10 mg/mL) was injected in each run, and a flow rate of 0.6 mg/mL (40 °C) was carried out. According to the standard curves of 8 standard substances (lgMw = −0.2028x + 12.709, *R*^2^ = 0.9929; lgMn = −0.1824x + 11.751, *R*^2^ = 0.9928) with specific molecular weights (5, 11.6, 23.8, 48.6, 80.9, 148, 273, 409.8, and 667.8 kDa), the Mw and Mn of NPPc was compared with the retention time.

### 2.4. Monosaccharide Composition of NPPc

Sample was hydrolyzed with 2.5 M trifluoroacetic acid (TFA) at 125 °C for 4 h, and then dried in a stream of air at 65 °C. Monosaccharide composition analysis was performed by the high performance anion-exchange chromatography (HPAEC, ThermoFisher ICS5000, Thermo Fisher Scientific, New York, NY, USA) on a CarbopacTMPA20 column (3 mm × 250 mm, Dionex) equipped with pulse-amperometric detection. Eluent contained A (0.015 M NaOH) and B (0.1 M NaOAc in 0.015 M NaOH). Monosaccharide standards were hydrolyzed with 2.5 M TFA at 125 °C for 4 h before they were applied to calibration.

### 2.5. Ultraviolet (UV) and FT-IR Spectrometry of NPPc

UV spectrum of the NPPc (0.5 mg/mL) was recorded in a spectrophotometer (Shimadzu UV-2550, Shimadzu Corporation, Kyoto, Japan) in the wavelength range of 190–400 nm. The infrared spectrum of NPPc was detected via the KBr disk method using a FI-IR spectrophotometer (Tianjin Port East Science and Technology Development Co., Ltd., Tianjin, China) at the wavelength range of 4000–400 cm^−1^. In a nutshell, 2.5 mg of NPPc sample dried by phosphorus pentoxide was mixed with 180 mg of KBr powder, and then pressed into a 1-mm-thick disk for analysis.

### 2.6. Methylation Analysis of NPPc

NPPc was methylated and analyzed by the gas chromatography-mass spectrometry (GC–MS) based on the previous method [22]. Approximately 30 mg of NPPc was methylated three times with methyl iodide. A complete methylation was monitored through the disappearance of O-H vibration absorption peak (3200–3700 cm^−1^) in IR spectrum. Permethylated samples were hydrolyzed with formic acid and TFA (2.5 M), and reduced with NaBH_4_ and acetylated with acetic anhydride. The reaction samples were injected to GC-MS system for GC–MS analysis (Shimadzu GCMS-QP 2010, Shimadzu, Kyoto, Japan). Chromatography was performed with the RXI-5 SIL MS column (30 m × 0.250 mm × 0.25 μm). The individual peaks of the partial methylated alditol acetates (PMAAs) were measured via the relative abundance of sugar residues and mass spectrum of PMAAs.

### 2.7. NMR Spectra Analysis of NPPc

NPPc (30 mg) was completely dissolved into 1 mL of D_2_O (99.9%) under constant stirring for 3 h, and then lyophilized. The deuterium-exchanged process was repeated three times. Subsequently, polysaccharides were re-dissolved in 0.6 mL D_2_O and filtered by 0.22 μm membranes, and then transferred into 5 mm NMR tube for testing [23]. The chemical shifts were provided in ppm, taking acetone-D_6_ signal (δ_H_ = 2.08 ppm) for high-resolution ^1^H spectrum, and acetone-D_6_ signal (δ_C_ = 30.39 ppm) for high-resolution ^13^C spectrum as internal references, respectively. Data analysis was operated with Bruker TopSpinTM program.

### 2.8. Cell Culture

The Caco-2 cells were incubated on 25 cm^2^ cassette culture dish in DMEM including 15% (*v*/*v*) of FBS and 1% (*v*/*v*) penicillin-streptomycin, and placed in the CO_2_ incubator. Complete culture medium was exchanged every other day and further grown until the cell fusion rate reached 75–90%. Subsequently, cells were passaged at a 1:3 split ratio by treating with 1 mL of 0.25% trypsin, digested for 5 min. The Caco-2 cells were used for all experiments.

### 2.9. Caco-2 Cells Uptake Experiment for NPPc

#### 2.9.1. Fluorescent Labeling of NPPc

FITC is widely used for protein labeling due to the high reactivity of isothiocyanate groups (N=C=S) with amino groups. Because NPPc does not contain amino groups, we introduced amino groups by adding tyramine. The reducing end (hemiacetal) of the polysaccharide easily reacts with the amino group to form a Schiff base, which is rapidly reduced to a stable secondary amine via nucleophilic addition reaction under the catalysis of NaBH_3_CN. The FITC derivative of PPc was labeled according to our previously described method [19].

#### 2.9.2. Cellular Uptake Quantitative Analysis by the Flow Cytometry Method (FCM)

Caco-2 cells were seeded at 2 × 10^5^ cells per well in 12 well plates and incubated in CO_2_ incubator prior to the uptake experiment. Then, 1 mL of 250, 500, 1000, 1500, and 2000 μg/mL the NPPc labeled by FITC (the same below) were added to each well and incubated for 2 h (or 1 mL of 1500 μg/mL NPPc labeled by FITC was added to each well and incubated for 1.0, 2.0, 3.0, and 4.0 h) to explore the effects of incubation concentration and time on cellular uptake. Next, the cells were washed two times with pre-cold PBS, trypsinized, and acquired in 0.5 mL HBSS. Supernatant was discarded by centrifugation at 3500× *g* for 10 min. Subsequently, cells were collected and re-suspended in 500 μL of HBSS before detection of cellular fluorescence intensity through FCM (BD FACSCalibur, Becton Dickinson, NJ, USA).

#### 2.9.3. Cellular Uptake and Localization by a Laser Scanning Confocal Microscopy

Sterilized coverslip was placed at the bottom of 12-well plate and 1 mL of 2 × 10^5^ Caco-2 cells was seeded on the coverslip to incubate in CO_2_ incubator for 24 h. Supernatant was discarded, and the NPPc (250, 500, 1000, 1500, and 2000 μg/mL) was added into 12-well plates and incubated with cells for different time intervals (0.5, 1.0, 2.0, 3.0, and 4.0 h). After incubation, Caco-2 cells were fixed by 4% paraformaldehyde at 37 °C for 10 min. Nucleus were stained with Hoechst 33,258 for 15 min (1:5000, diluted by PBS), and mitochondria and the endoplasmic reticulum were stained with Mito Tracker Red for 60 min and ER-Tracker Red for 30 min, respectively. Images were tested by a TCS SP5II laser scanning confocal microscopy (Leica, Weztlar, Germany).

### 2.10. Data Statistical and Analysis

All data were presented as the mean value ± SD achieved by at least three independent experiments. The statistical analysis was performed by GraphPad Prism 8.00.

## 3. Results

### 3.1. Preparation of NPPc and Detection of Molecular Weight

The NPPc was acquired from *Cucurbia moschata* by hot-water extraction and a stepwise ethanol fractionated precipitation. As displayed in Figure 1A, the elution peak of the polysaccharide from the HPGPC is single and symmetric which reflected the trait of homogeneous distribution. The total yield of NPPc was closed to 1.03% (lyophilized weight, w/w) based on fresh weight of raw material. The UV-vis spectrum (Figure 1B) indicated that NPPc had no obvious absorption peaks at 260 and 280 nm, proving the absence of proteins and nucleic acids of NPPc. Based on the equation of the calibration curve of dextran, the average molecular weight (Mw) and number average molecular weight (Mn) of NPPc were 9023 and 7549 Da, respectively. Furthermore, polydispersity index of four polysaccharides (Mw/Mn) was 1.20, which means that NPPc had a narrow molar mass distribution and the molecule existed in a less dispersed form in aqueous solution. Compared with the column fractionation, the stepwise ethanol fractionated precipitation was a feasible method to obtain the polysaccharide with a specific molecular weight.

### 3.2. Monosaccharide Composition

The sugar composition analysis of NPPc was obtained by comparing the retention times to standards by HPAEC (Figure 2). Results indicated that NPPc contained L-arabinose, D-galactose, and D-glucose—in a molar ratio of 0.9:1.7:97.4, suggesting that it was a heteropolysaccharide consisted of different monosaccharide composition, particularly, D-glucose was a main monosaccharide.

### 3.3. FT-IR Spectrum Characteristics

The FT-IR spectrum (Figure 3) was employed to identify the functional groups of NPPc through representative absorption peaks. Primarily, a wide and strong peak at 3363.25 cm^−1^, and a sharp peak at 2927.41 cm^−1^ were attributed to the presence of O-H, and C-H stretching vibrations, respectively [24,25]. The absorption band at around 1635.34 cm^−1^ may be caused by the absorbed water, implying strong glucan-water hydrogen interaction [26]. The band appearing at 1421.28 cm^−1^ was derived from the antisymmetric and symmetric vibrations of the C-H in carboxyl group [27]. The bands in the region from 1200 to 1000 cm^−1^ were attributed to stretch vibrations of C-O-C and C-O-H side groups, implying the presence of pyranose ring [28]. The absorption bands at 862 and 761 cm^−1^ confirmed that NPPc was D-glucopyranose derivatives. Furthermore, the weak bands near 761.74 cm^−1^ were assigned to ring stretching and ring deformation of α-d-(1–4) linkages [29]. Eventually, FT-IR spectrum of NPPc displayed the typical absorption peaks of polysaccharides.

### 3.4. Methylation Analysis

Methylation analysis was used to determine the glycosyl linkage types. NPPc in the current study indicated four peaks in the total ion chromatogram, which were attributed to terminal Glc*p*, 1,4-linked Glc*p*, 1,6-linked Glc*p*, and 1,4,6-linked Glc*p* residues by comparing with the relative retention time and mass spectrum of the partial methylated alditol acetates (PMAAs) (Table 1). Each glycosidic linkages pattern has its own specific characteristic ion fragments. Retention times of standard PMAAs were also used to identify the glycosidic linkages pattern. The corresponding molar ratio of the four linkages in NPPc was about 9.8: 70.1: 13: 7.1, clarifying the backbone chain of (1→4)-Glc*p* residue with the branch at *O*-6 position. The above results did not provide relevant information on Ara*f* and Gal*p* residues, most likely due to their very low content accounted for only about 2% of the monosaccharide analysis [30]. In addition, the molar ratio between terminal units and the branched points was 1.3: 1.0. Structural characteristics of NPPc needed to be in-depth confirmed through 1D and 2D NMR spectra.

### 3.5. NMR Spectra Analysis

According to the results of monosaccharide composition and methylation analysis, the structure of NPPc was further interpreted via 1D-NMR (^1^H- and ^13^C- NMR spectra) and 2D-NMR (COSY, HSQC, and HMBC spectra) [31]. According to ^1^H NMR spectrum, six anomeric dominant proton signals at δ 5.29, 4.87, 5.25, 4.86, 5.11, and 4.53 ppm were assigned to residues A, B, C, D, R_α_, and R_β_, respectively. Corresponding, these six anomeric carbon signals at δ 99.61, 97.68, 99.71, 97.67, 91.67, and 95.63 ppm were determined by ^13^C NMR and HSQC spectra. In this study, the summarized results of the NMR spectra analysis were showed in Table 2 and Figure 4.

There were no characteristic signals of Gal*p* and Ara*f* residues in the NMR spectra of NPPc, because the monosaccharide content of two sugars were small (2%), less than 5%. Moreover, there was no carboxyl signal at δ 160–180 ppm in the ^13^C NMR spectrum, demonstrating that the absence of uronic acid in NPPc, which was in a good consistent with the FT-IR spectrum.

For residue A, the anomeric chemical shift at δ 5.29 ppm manifested that the residue A may be α-configuration unit. In the ^1^H-^1^H COSY spectrum (Figure 4C), cross peaks at δ 5.29/3.46, δ 3.46/3.86, δ 3.86/3.54, and δ 3.54/3.30 ppm were observed, which were attributed to H-1/H-2, H-2/H-3, H-3/H-4, and H-4/H-5 signals. R_α_ and R_β_ signals can be attributed as δ 3.62 and δ 3.73 ppm, respectively, by the correlation spectrum of HSQC (Figure 4D). Meanwhile, the corresponding carbon signal appeared from the HSQC spectrum (Figure 4D) were δ 99.61, 71.50, 73.30, 76.81, 69.47, and 60.42 ppm for C-1 to C-6, respectively. All the ^1^H and ^13^C chemical shifts of residue A were in accordance with the literature reports [32] and chemical shifts of C-1 and C-4 was shifted to low magnetic field, demonstrating that the substitution of residue A occurred at *O*-1 and *O*-4. Therefore, residue A was confirmed to be →4)-α-D-Glc*p*-(1→. Combining the above analysis, methylation analysis and previous reported literature [32,33,34,35,36,37], similar to residue A, residue B, C, D, R_α_, and R_β_ were identified as →6)-α-D-Glc*p*-(1→, →4,6)-α-D-Glc*p*-(1→, α-D-Glc*p*-(1→, →4)-α-D-Glc*p*, and →4)-β-D-Glc*p*, respectively. The integrated assignments of all ^1^H and ^13^C NMR chemical shifts are listed in Table 2. Additionally, the ratio of alpha-1,4/alpha-1,6 could be calculated as 3.12: 1 using the signal of anomeric proton of →4-α-D-Glc*p*-(1→ (δ 5.29 ppm) and →6)-α-D-Glc*p*-(1→ (δ 4.87 ppm) peak area ratios in the ^1^H-NMR spectrum.

The glycosidic linkage sequence among the NPPc and both intra- and inter-residual correlations were determined according to the correlation peaks acquired in the HMBC spectrum (Figure 4E). A strong correlation peak at δ 5.29/76.81 ppm confirmed the correlation between H-4 and C-1 of residue A, illustrating a (1→4)-α-D-linked glucan backbone. A cross-peak signal at δ 5.29/78.01 ppm was attributed to the correlation between the H-1 (δ 5.29) of residue A and the C-1 (δ 78.01) of residue C, indicating that the H-1 of residue A is linked to the C-1 of residue C. Another cross peak at 4.86/65.36 ppm (H-1 of residue D and C-6 of residue B) indicated that the side chain Glc*p* is linked to Glc*p* at C-6 position. The H-1of →6)-α-D-Glc*p*-(1→ was attached to C-6 of →6)-α-D-Glc*p*-(1→ as indicated by a cross peak at δ 4.87/65.36 ppm in the HMBC spectrum. The cross peak δ 4.87/65.59 ppm showed the correlation between H-1 of →6)-α-D-Glc*p*-(1→ and C-6 of →4,6)-α-D-Glc*p*-(1→. Furthermore, residuals R_α_ and R_β_ did not show cross signal in HMBC correlation spectrum, which may be due to the low monosaccharide content. Considering the results of methylation analysis, the molar ratio of residues A, B and C was about 10: 2: 1. Therefore, a hypothetical structure for NPPc was proposed in Figure 4F according to the comprehensive results of monosaccharide composition, methylation analysis, and the 1D and 2D NMR.

### 3.6. Cellular Uptake of NPPc by Caco-2 Cell

The cellular uptake of NPPc on Caco-2 cells was investigated and incubated with NPPc at a predetermined time and concentration by the laser confocal spectrum microscope method and FCM. As displayed in Figure 5 and Figure 6, fluorescence intensity changed from weak to strong with the extension of concentration and incubation time, which was consistent with flow cytometry analysis. When NPPc incubated concentration was 250–500 μg/mL and incubated time was 0.5 h, a small amount of cells were detectable for NPPc green fluorescence, supposing that a few NPPc had been uptaken by cells. However, when the concentration ranged from 1000 to 2000 μg/mL within 4 h, a strong green fluorescence signal was observed. In addition, the cell nucleus of Caco-2 cells was surrounded by NPPc and no green signals were found within the nucleus. Uptake studies showed that NPPc entered into Caco-2 cells was time- and concentration-dependent. Dou et al. [38] incubated the nanoparticle modified chitosan with Caco-2 cells to investigate its cellular uptake. The results also showed that the fluorescence intensity varied from weak to strong with the increase of modified nanoparticles concentration, which was consistent with our study.

### 3.7. Subcellular location of NPPc in the Caco-2 Cell

The amount of NPPc in bilateral sides was lower than the initial addition. Hence, we speculated that the reduced portion might be ingested by cells. Our previous research has reported that pumpkin polysaccharides have a range of cellular active functions [6,9]. To date, it is unclear yet whether pumpkin polysaccharides exert their bioactivities by binding to cell surface receptors or entering the intracellular environment through endocytosis. Hence, studying the subcellular localization of pumpkin polysaccharides may help us to further understand the functional mechanisms of pumpkin polysaccharides [39]. In the current study, NPPc was selected as a representative of pumpkin polysaccharide to detect its subcellular localization in Caco-2 cells by using the organelle specific labeling.

It is well known that a wide variety of organelles were distributed in the cytoplasm. According to Figure 5, NPPc was diffusely distributed in Caco-2 cells, suggesting that NPPc might be located in some intracellular organelles. Hence, the positional relationships between NPPc (1500 μg/mL) and two major organelles (mitochondria and endoplasmic reticulum) were intensively investigated. As shown in Figure 7A,B, green and red fluorescence obviously overlapped and orange fluorescence signal appeared, indicating that NPPc was located in mitochondria and endoplasmic reticulum within 240 min. Fluorescence intensity of NPPc in the mitochondrion and endoplasmic reticulum increased with time points, suggesting that NPPc may be transported to the mitochondrion and endoplasmic reticulum to exert bioactivities.

For example, it has been reported that the mice administrated with neutral *Hohenbuehelia Serotina* polysaccharides (NTHSP) could significantly inhibit the splenocytes apoptosis induced by γ-radiation by blocking the endoplasmic reticulum apoptosis pathway: PERK-ATF4-CHOP, IRE1 alpha-XBP1-CHOP, and ATF6-XBP1-CHOP [40]. Shen et al. [41] reported that PPW-induced inhibition of cell proliferation in HepG2 cells was associated with the induction of apoptosis. Exposure of HepG2 cells to PPW (100, 200, and 400 μg/mL) resulted in a loss of mitochondrial membrane potential (Δψm) and the release of cytochrome c from the mitochondria to the cytosol. Those studies of NTHSP and PPW subcellular localization provided the basis for a more detailed analysis of the interaction with cells, which helped to study the mechanisms of the effects of pumpkin polysaccharides on various cells. Moreover, it was detected that NPPc is located not only in mitochondria and endoplasmic reticulum, but also in other organelles during 180–240 min.

## 4. Discussion

In the current study, primary structure characterization on a polysaccharide from *Curcubita monstacha* was investigated at first. NPPc, was successfully obtained and purified by the gradient ethanol precipitation method. The chemical structure properties confirmed that NPPc mainly consists of D-glucose with average molecular weight of 9.023 kDa. According to the analysis of FT-IR spectrum, methylation analysis and NMR spectra, NPPc was proposed to be composed of a main chain of (1→4)-linked-α-D-Glc*p* and the branch of (1→4,6)-linked-α-D-Glc*p* substituted at *O*-6. We found that polysaccharides reported in some literatures [2,32,35,36,42] are all mainly comprised of a linear repeating backbone of (1→4)–linked-α-D-Glc*p* residues and a branched chain, which are similar to the structure of NPPc, and have a strong immunomodulatory effect. These investigations provided the guidance for our further study on bio-functions of the NPPc. Furthermore, uptake characteristics of NPPc in Caco-2 cells were evaluated. The subcellular location of NPPc in mitochondrion and endoplasmic reticulum were detected. Next, the Caco-2 cells monolayer model needs to be established to explore the absorption of intestinal epithelium characteristics and mechanisms of pumpkin polysaccharides, and performed in animals experiments to further verify the results of the cells experiment mentioned above.

## Figures and Tables

**Figure 1 foods-10-02357-f001:**
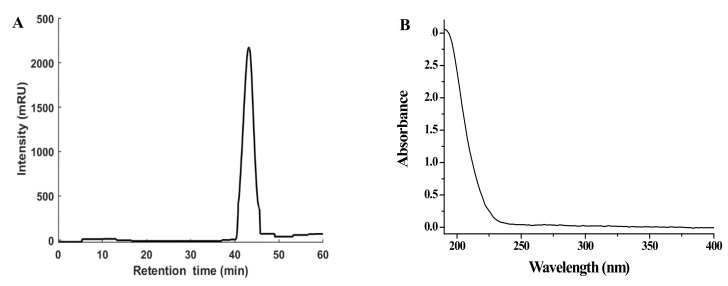
High performance gel permeation chromatography (HPGPC) of NPPc (**A**) and UV–vis spectrum of NPPc (**B**).

**Figure 2 foods-10-02357-f002:**
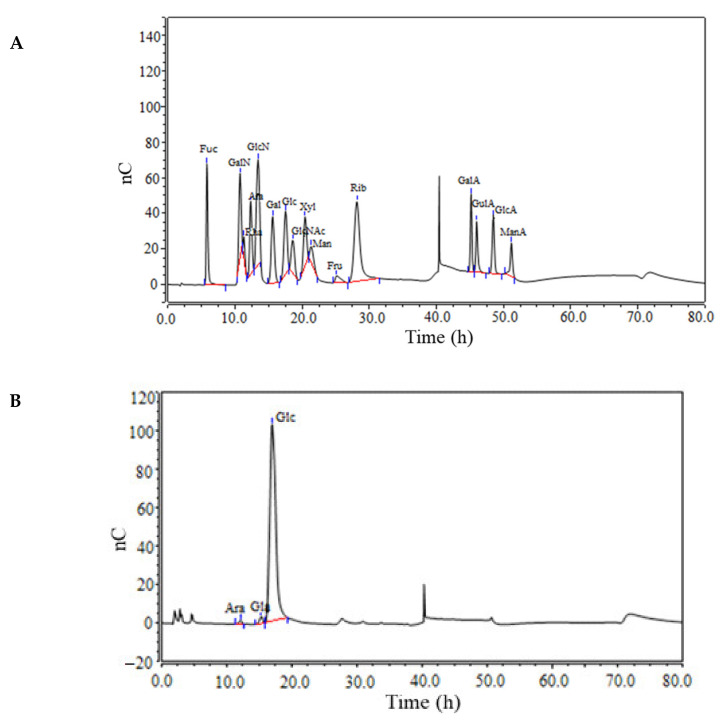
Monosaccharide composition of NPPc. Chromatograms of standard mixture of 15 monosaccharides (**A**) and the composed monosaccharides in NPPc (**B**).

**Figure 3 foods-10-02357-f003:**
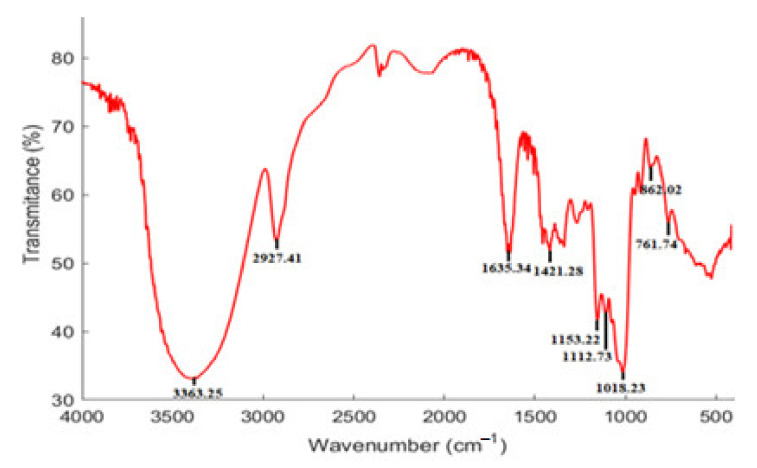
FT-IR spectrum of NPPc.

**Figure 4 foods-10-02357-f004:**
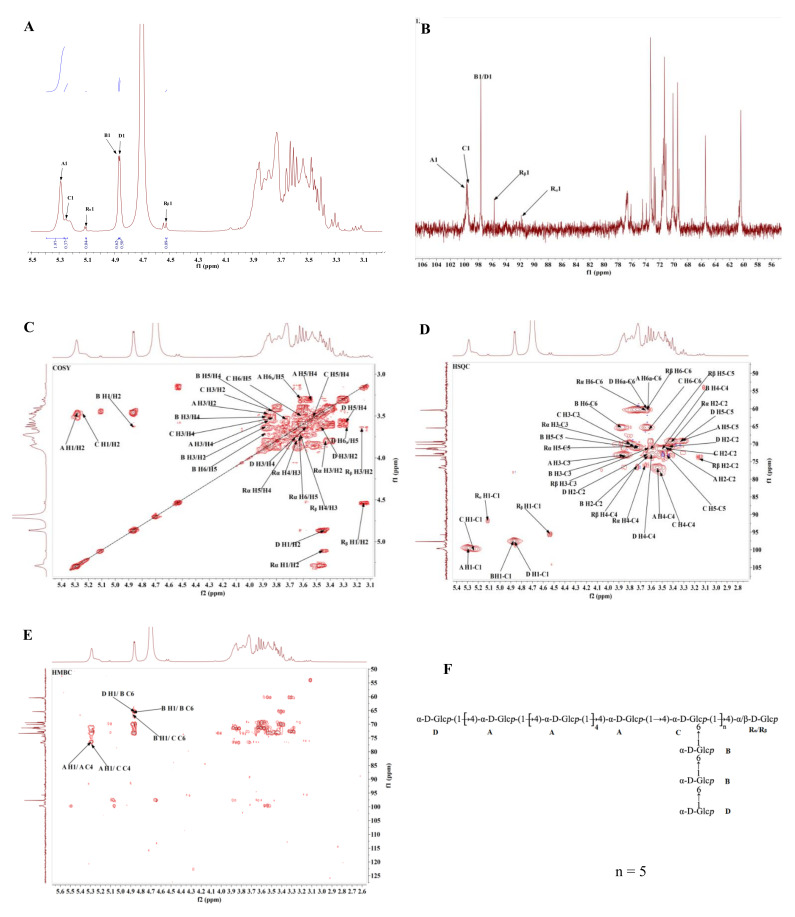
^1^H spectrum (**A**), ^13^C spectrum (**B**), COSY spectrum (**C**), HSQC spectrum (**D**), HMBC spectrum (**E**), and predicted repeating unit of NPPc (**F**).

**Figure 5 foods-10-02357-f005:**
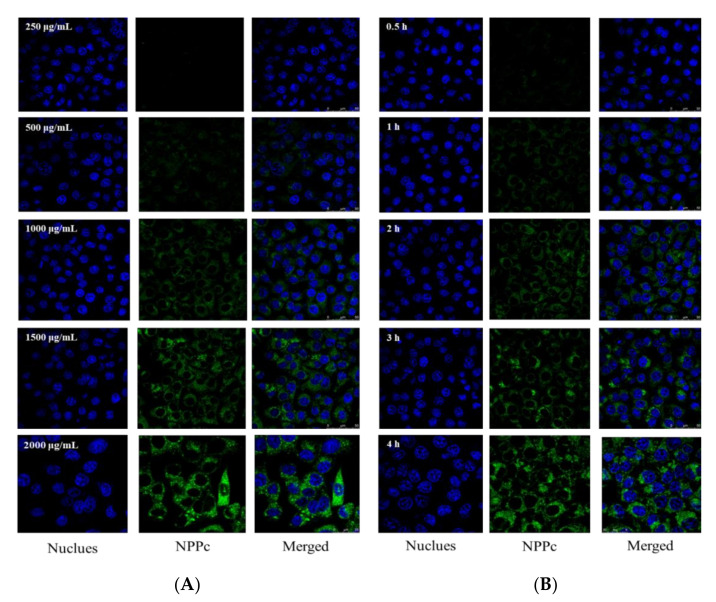
Localizations of NPPc in Caco-2 cells with various concentrations (**A**) and time intervals (**B**) by a laser scanning fluorescence microscope.

**Figure 6 foods-10-02357-f006:**
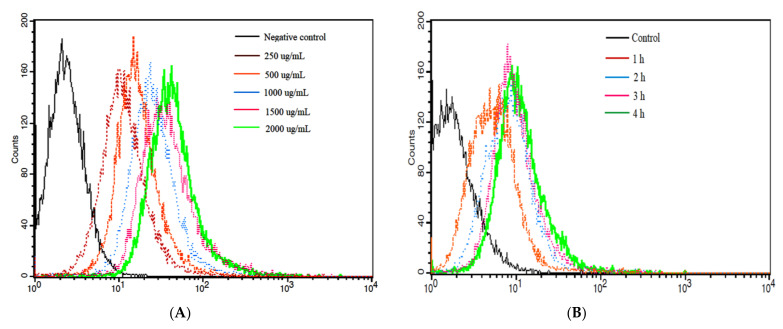
The internalized NPPc in Caco-2 cells via flow cytometry method (FCM). Effects of a series of incubation concentrations on the uptake of NPPc for 4 h (**A**), and effects of a serious of incubation time points on the uptake of NPPc at a concentration of 500 μg/mL (**B**).

**Figure 7 foods-10-02357-f007:**
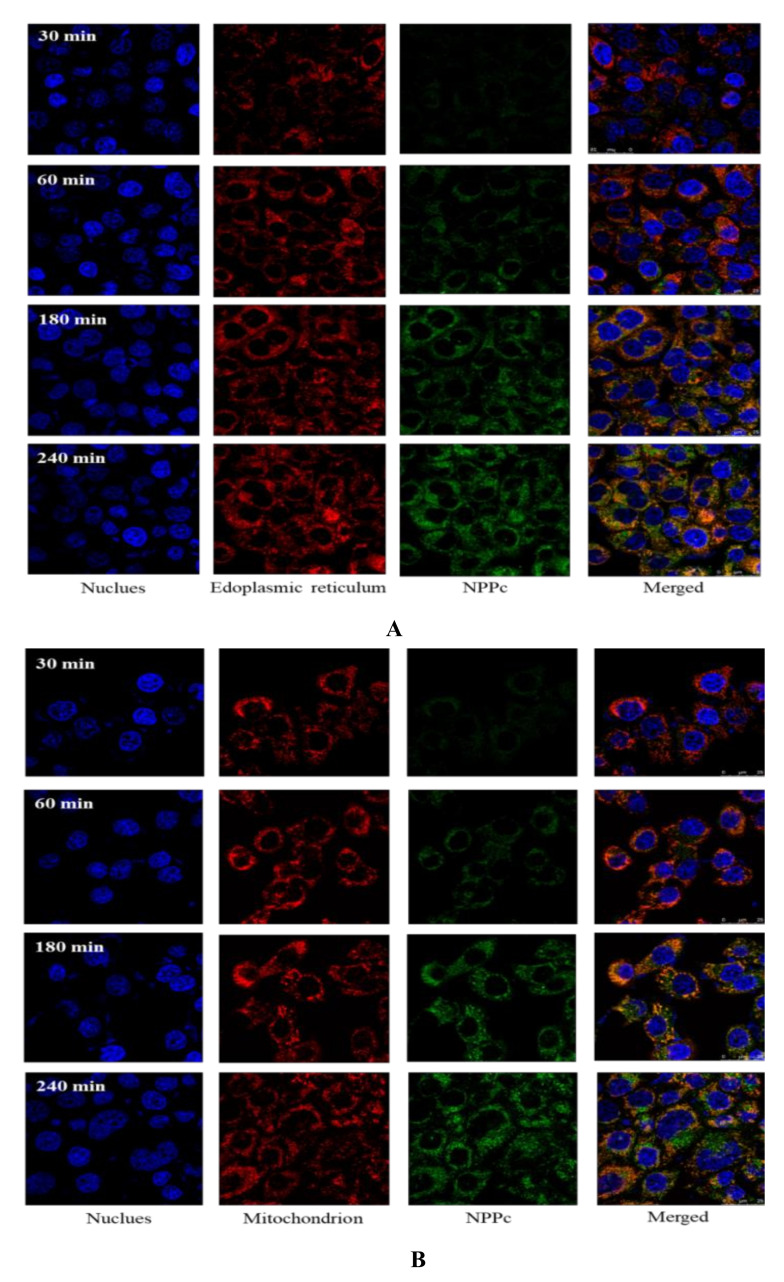
Localization of NPPc (1500 μg/mL) in endoplasmic reticulum (**A**) and mitochondrion (**B**).

**Table 1 foods-10-02357-t001:** Methylation analysis of NPPc.

Retention Time	Methylated Sugar	Mass Fragments (m/z)	Molar Ratio	Types of Linkage
16.182	2,3,4,6-Me_4_-Glc*p*	43,71,87,101,117,129,145,161,205	9.7	Glc*p*-(1→
21.755	2,3,6-Me_3_-Glc*p*	43,87,99,101,113,117,129,131,161,173,233	70.1	→4)-Glc*p*-(1→
22.425	2,3,4-Me_3_-Glc*p*	43,87,99,101,117,129,161,189,233	13	→6-Glc*p*-(1→
27.255	2,3-Me_2_-Glc*p*	43,71,85,87,99,101,117,127,159,161,201	7.2	→4,6)-Glc*p*-(1→

**Table 2 foods-10-02357-t002:** ^1^H and ^13^C NMR chemical shifts (ppm) of NPPc in D_2_O.

Glycosyl Residues		Chemical Shift δ H/C (ppm)							
			1	2	3	4	5	6a	6b
A	→4-α-D-Glc*p*-(1→	H	5.29	3.46	3.86	3.54	3.30	3.62	3.73
		C	99.61	71.50	73.30	76.81	69.47	60.42	
B	→6)-α-D-Glcp-(1→	H	4.87	3.61	3.87	3.51	3.71	3.87	
		C	97.68	72.90	73.44	71.40	70.80	65.36	
C	→4,6)-α-D-Glc*p*-(1→	H	5.25	3.44	3.82	3.51	3.48	3.63	
		C	99.71	71.64	70.07	78.01	73.07	65.59	
D	α-D-Glc*p*-(1→	H	4.86	3.45	3.62	3.58	3.31	3.63	3.73
		C	97.67	71.27	73.07	73.07	69.23	60.44	
R_α_	→4)-α-D-Glc*p*	H	5.11	3.43	3.78	3.64	3.72	3.62	3.71
		C	91.76	69.33	70.20	75.97	71.14	60.56	
R_β_	→4)-β-D-Glc*p*	H	4.53	3.16	3.63	3.73	3.57	3.62	3.73
		C	95.63	73.81	71.67	76.58	71.74	60.61	

## Data Availability

Not applicable.

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
