# Peer review of "Structural Characterization of a Neutral Polysaccharide from Cucurbia moschata and Its Uptake Behaviors in Caco-2 Cells"

_foods, 2021, doi:10.3390/foods10102357_

Round 1
Reviewer 1 Report
The title of “structural characterization of a neutral polysaccharide from cucurbia mischata and its uptake behavior in Caco-2cells” was evaluated the structural properties of pumpkin polysaccharide with using various instruments and studied its cellular uptake behavior using Caco-2 cells. Although, this study is interesting to investigate the molecular structure of pumpkin polysaccharide and cell work with macromolecules, this article requires intensive polishing of article and improve the writing to publish on “Foods”. The article contained a lot of grammatical and other small errors; thus it was difficult to smooth reading.
Also, its study covered 1) structural characteristics 2) cell study. But, both studies described very roughly. Even you choose one study, please provide more in-depth results and discussion. Also, you mention about the minor contamination of gal and ara. Where those are come from? Where and how those are contaminated. This is very important point of your study. Study carried out triplicated as you write on. Is it happened all three times? Please verify this contamination logically.
- Errors and suggestions. L23-24 etc -à please rewrite the sentenceL34 Chen et al. -à please added the reference after that.L65. FTIC--à please provide the full nameL74. Tyramine --à I couldn’t find the tyramine use in the other part of article. Where are you used?L90-91. 65% solution directly carry out the centrifugation? Any washing step of 65% ethanol solution?L192. Mainly-à main?L195-197--à logically weird, please rewrite.L272. Fig. 5F ??? -à 4FL300. Serious ???---àserial ???L320. Need to put reference. Fig 2. Fig3. Could you redraw instead using original image?
- L 367. No one carry out the experiments?
- L305. My previous -à rewrite.
- Fig. 4F--à branch point is that right alpha-D-6Glcp?
- L243-245, L252-253-à need to be improve the sentence.
- L192. I think-à please rewording. I think
- L173. Rewrite the sentence
- L71-165. Instrument or reagent (company, city, country)--à please corresponding.
- L72. Seeded --à may be seed out?
- L39. Later -à the other study?
- L26. Of the vegetable--à sentence is not meaningful
- L 12. NPPC-à NPPc
- Others.L221. You mention about the minor contamination of gal and ara. Where those are come from. Where and how those are contaminated. This is very important point of your study. You must logically persuade the readers about this issue. Study carried out triplicated as you write on. Is it happened all three times?.Results 3.5 L324-330. Discussion need to be rewrote with logically. L348-354 seems rewrite. The session is too far from your study.
- Discussion
- I’m not quite sure those are relevant reference of Caco-2 cell study?. Please provide more detail description. Also need to background information of Caco-2 cell.
- Result 3.7
- Also, if you provide the schematic/3D-like (or more extended scheme) visual polysaccharide you determined, it should be good to understand. It looks somewhat similar to starch, so if you compare pumpkin polysaccharide to starch will be great. Also, please provide the branching ratio of alph-1, 4/ alpha-1,6 with 1HNMR study. 1HNMR is able to calculate the ratio of (1,4/1,6)
- About the results of your experiments, the pumpkin polysaccharide resulted in alpha-glucan. Alpha-1, 4 glucose backbone and branch point of alpha (1, 6)-glucose. So, Is it similar to starch?. in abstract you mentioned the side chain contained (1, 6) beta-D-glucose, which indicate the side chain (1, 6) beta-D-glucose on Table 2. Results 3.5 is hard to follow up. How about you indicate/point out the A, B, C, D on Fig 4F.
- *. L57. Please provide the more information of Caco-2 cells. The cell line for experiments with macromolecules such as polysaccharide is not familiar with many general readers. So, please provide more information and previous study of Caco-2 cells either introduction or results.
-
L221. You mention about the minor contamination of gal and ara. Where those are come from. Where and how those are contaminated. This is very important point of your study. You must logically persuade the readers about this issue. Study carried out triplicated as you write on. Is it happened all three times?
Results 3.5
About the results of your experiments, the pumpkin polysaccharide resulted in alpha-glucan. Alpha-1, 4 glucose backbone and branch point of alpha (1, 6)-glucose. So, Is it similar to starch?. in abstract you mentioned the side chain contained (1, 6) beta-D-glucose, which indicate the side chain (1, 6) beta-D-glucose on Table 2. Results 3.5 is hard to follow up. How about you indicate/point out the A, B, C, D on Fig 4F.
Also, if you provide the schematic/3D-like (or more extended scheme) visual polysaccharide you determined, it should be good to understand. It looks somewhat similar to starch, so if you compare pumpkin polysaccharide to starch will be great. Also, please provide the branching ratio of alph-1, 4/ alpha-1,6 with 1HNMR study. 1HNMR is able to calculate the ratio of (1,4/1,6)
Result 3.7
L324-330.
I’m not quite sure those are relevant reference of Caco-2 cell study?. Please provide more detail description. Also need to background information of Caco-2 cell.
Discussion
Discussion need to be rewrote with logically. L348-354 seems rewrite. The session is too far from your study.

Reviewer 2 Report
In this study, the authors identified the structure of a polysaccharide extracted from a pumpkin and investigated the cellular uptake kinetics using Caco-2 cell line. The authors did plenty of work on the structural characterization of this polysaccharide, including monosaccharide composition, molecular weight by PGC, linkage by permethylation and GCMS, detail structure by NMR, along with additional UV and IR analysis. It covered all necessary aspects for a polysaccharide. The investigation on the cellular uptake is a plus for this manuscript. Thus, it could be a very valuable addition to the literature of plant carbohydrate. However, there are some major problems, and the language must be improved. I suggest resubmitting this manuscript after a complete and careful revision.
Major problems:
- Page4, line 151-154, “Next, the cells were washed two times with pre-cold PBS, and trypsinized and acquired in 0.5 mL HBSS. Supernatant was obtained by centrifugation at 5000 rpm/min for 10 min. Then, cells were collected and re-suspended in 500 μL of HBSS before detection of cellular fluorescence intensity through FCM”. Why did the authors obtain the supernatant in the second sentence? Should it be discarded? Was 5000 rpm/min too fast to kill the cells?
- “Molecular mass” should be “molecular weight”. Additionally, even though the GPC shows a single peak, the polysaccharide sample extracted from plants is usually considered a mixture of polysaccharides with a small range of degree of polymerizations (DP) and slightly different structure. Thus, it is suggested to use “average molecular weight”. Moreover, the authors need to consider what kind of “average molecular weight” they are reporting? Mn, Mw, Mz?
- In the results of methylation analysis using GC-MS, the authors indicated that the determination of residues “by comparing with the literature”. It is unclear how the authors differentiated the structure of Glc and that of other hexoses, e.g. Gal, since they have the same mass for the same structure? For example, how to differentiate “2,3,4,6-Me4-Glcp” to “2,3,4,6-Me4-Galp”? by retention time? Will “2,3,4,6-Me4-Glcp” elute at the same time with other “2,3,4,6-Me4-Hexp”? please list the retention times of the major analogues or add one or two sentences to address this issue. Additionally, the absence of Ara and Gal signals may just due to their small proportions in the polysaccharide which did not give enough signal in GC-MS.
- In line 235-239, the authors claimed that the absence of Gal and Ara signals “proves our speculation that Galp and Araf are contaminants”. However, NMR will not give signals since they are <5%. The absence of signals in NMR is not the evidence to support that conclusion.
- The concluded structure is inconsistent in the Abstract and Fig 4F. In the Abstract, “The side chain contained (1→6)-β-D-Glcp and terminal glucose” while in the predicted structure in Fig.4F, the side chain only contains (1→6)--α-D-Glcp and terminal glucose.
- The authors think Gal and Ara are contaminants of NPPc. Where is the contamination from? Fail to remove them during the extraction and purification or it is simply a contamination during the analysis of monosaccharide composition? If just contaminants during the monosaccharide composition analysis, why don’t the authors do that analysis again and obtain a result without Gal and Ara? If the authors think Gal and Ara are contaminants, more solid evidence is needed. The ambiguity in the structure and the explanation of contaminant of Gal and Ara is the major reason of rejection.
- Can GPC determine the molecular weight as accurate as 9.023 kDa or 9023 Da? 9 kDa or 9.0 kDa will be more acceptable.
Minor issues:
- It lacks the short introduction of Caco-2 cell line where it pops up in the Introduction section. It is unclear for most of the readers why this cell line can be used to “simulate intestinal epithelial cells”. It is suggested that the authors either inform “human colorectal adenocarcinoma cells” or “It is primarily used as a model of the intestinal epithelial barrier”
- I don’t understand why the authors made a comparison in line 193-197 since neither the materials, methods, nor the results are comparable.
- “Monosaccharide compositions” should be “monosaccharide composition”
- 2, please use a larger font to mark the peaks. “Chromatograms of 15 mixed monosaccharide standards” may be revised to “chromatograms of standard mixture of 15 monosaccharides”. “the composed monosaccharides (B) in NPPc” should be “the composed monosaccharides in NPPc (B)”
- Page5, line 192, “D-glucose was a mianly monosaccharide”. “mianly” might be a typo of “mainly”. It might be better to use “predominant”. In the same line, “Notably, I think the low amounts of galactose and arabinose are small contaminants”. I have never seen “I think” in a manuscript with multiple authors. Who is the “I” that made the “think”?
- Line 194, “included” should be “contained”.
Round 2
Reviewer 1 Report
The submitted revised MN has been significantly modified followed by reviewers suggestion. Now, the MN looks more sound and strong article than 1st version. However, it contained a lot of small grammatical and text errors. English editing is strongly required.
L25 Many on?
Response 1: Line 25: Thanks for the reviewer’s careful check. I apologize for our clerical error. According to the reviewer suggested, we have removed “on” in the revised manuscript. Meanwhile, the revised manuscript has been checked by a native English-speaking colleague. Please check it out in the revised manuscript.
L36 read again
Response 2: Line 36: Thanks for the reviewer’s careful check. We have revised the sentence according to the reviewer’s advice. The sentence has been changed to “Chen et al. [15] obtained a neutral pumpkin polysaccharide extracted by aqueous two-phase system, which was constituted by (1→3)-linked-Glcp as backbone.” Please check it out in the manuscript.
L37 mentioned the in their study ?
Response 3: Line 37: Thanks for the reviewer’s careful check. We have revised the sentence according to the reviewer’s advice. The sentence has been changed to “Unfortunately, pumpkin species was not mentioned in their study.” Please check it out in the manuscript.
L456
Response 4: Line 456: Thanks for the reviewer’s careful check. We have revised the sentence according to the reviewer’s advice. The sentence has been changed to “Additionally, the ratio of alpha-1,4/ alpha-1,6 could be calculated as 3.12: 1 using the signal of anomeric proton of →4-α-D-Glcp-(1→ (δ 5.29 ppm) and →6)-α-D-Glcp -(1→ (δ 4.87 ppm) peak area ratios in the 1H-NMR spectrum.” Please check it out in the manuscript.
L459
Response 5: Line 459: Thanks for the reviewer’s suggestion. We have edited this sentence and changed “Fig. 44E” to “Fig. 4E” in the revised manuscript. Please check it out in the revised manuscript.
L615 I couldn't find the replaced fig 7.
Response 6: Thanks for the reviewer’s careful check. We have rechecked this problem in the Word version, please check whether could find in the PDF version.
L619 Even though I suggested "starch-like" structure for NPPc, it is too much jump out the conclusion. specially branch contained alpha 1->6 linkage. please found another description or delete the term.
Response 7: Line 619: Thanks for the reviewer’s suggestion. We agreed with the reviewer’s comment and removed the sentence of “In the current study, we called the NPPc “Starch-like” polysaccharide.” according to the reviewer’s advice. Please check it out in the revised manuscript.
Author Response
Point 1:L25 Many on ?
Response 1: Line 25: Thanks for the reviewer’s careful check. I apologize for our clerical error. According to the reviewer suggested, we have removed “on” in the revised manuscript. Meanwhile, the revised manuscript has been checked by a native English-speaking colleague. Please check it out in the revised manuscript.
Point 2: L36 read again
Response 2: Line 36: Thanks for the reviewer’s careful check. We have revised the sentence according to the reviewer’s advice. The sentence has been changed to “Chen et al. [15] obtained a neutral pumpkin polysaccharide extracted by aqueous two-phase system, which was constituted by (1→3)-linked-Glcp as backbone.” Please check it out in the manuscript.
Point 3: L37 mentioned the in their study ?
Response 3: Line 37: Thanks for the reviewer’s careful check. We have revised the sentence according to the reviewer’s advice. The sentence has been changed to “Unfortunately, pumpkin species was not mentioned in their study.” Please check it out in the manuscript.
Point 4: L 456
Response 4: Line 456: Thanks for the reviewer’s careful check. We have revised the sentence according to the reviewer’s advice. The sentence has been changed to “Additionally, the ratio of alpha-1,4/ alpha-1,6 could be calculated as 3.12: 1 using the signal of anomeric proton of →4-α-D-Glcp-(1→ (δ 5.29 ppm) and →6)-α-D-Glcp -(1→ (δ 4.87 ppm) peak area ratios in the 1H-NMR spectrum.” Please check it out in the manuscript.
Point 5: L459
Response 5: Line 459: Thanks for the reviewer’s suggestion. We have edited this sentence and changed “Fig. 44E” to “Fig. 4E” in the revised manuscript. Please check it out in the revised manuscript.
Point 6: L615 I couldn't find the replaced fig 7.
Response 6: Thanks for the reviewer’s careful check. We have rechecked this problem in the Word version, please check whether could find in the PDF version.
Point 7: L619 Even though I suggested "starch-like" structure for NPPc, it is too much jump out the conclusion. specially branch contained alpha 1->6 linkage. please found another description or delete the term.
Response 7: Line 619: Thanks for the reviewer’s suggestion. We agreed with the reviewer’s comment and removed the sentence of “In the current study, we called the NPPc “Starch-like” polysaccharide.” according to the reviewer’s advice. Please check it out in the revised manuscript.

Reviewer 2 Report
The authors improved the manuscript greatly in this version. The language is generally acceptable now but with some typos. Most of my previous concerns were addressed except the "methylation analysis" as describe below. However, with so many changes, some contents are hard to read. For example, is Figure 4F deleted or not? A clear version may help to make the decision.
Small issues:
- Description of methylation analysis is still ambiguous. As the authors replied, "Each glycosidic linkages pattern has its own specific characteristic ion fragments and the retention time. " This is how the analogues are differentiated. Please put this sentence in the paragraph and add one or two sentences to make the readers clear. The majority of the readers may be out of the field of carbohydrate analysis and may not understand without extra explaination.
- line 204, "hydrolyzised" should be "hydrolyzed". Same below in line 210.
- In this paragraph, the authors need to give abbv "TFA" in line 204 so that they can directly use "TFA" in line 210.
- line 364, "cuased"
- My previous concern on the digit of Mw was whether the Mw determined by HPGPC can be as accurate as 10 Da or not. please keep using 9023Da along with 7549 Da.
Author Response
Point 1: Description of methylation analysis is still ambiguous. As the authors replied, "Each glycosidic linkages pattern has its own specific characteristic ion fragments and the retention time. " This is how the analogues are differentiated. Please put this sentence in the paragraph and add one or two sentences to make the readers clear. The majority of the readers may be out of the field of carbohydrate analysis and may not understand without extra explaination.
Response 1: Thanks for the reviewer’s suggestion. As the reviewer suggested, we have added the explanations in Results 3.4 according to the reviewer’s advice. In detail, each glycosidic linkages pattern has its own specific characteristic ion fragments. Retention times of standard PMAAs were also used to identify the glycosidic linkages pattern. Please check it out in the revised manuscript.
Point 2: line 204, "hydrolyzised" should be "hydrolyzed". Same below in line 210.
Response 2: Line 210: Thanks for the reviewer’s careful check. We have changed “hydrolyzised” to “hydrolyzed” in the revised manuscript. Please check it out in the revised manuscript.
Point 3: In this paragraph, the authors need to give abbv "TFA" in line 204 so that they can directly use "TFA" in line 210.
Response 3: Line 210: Thanks for the reviewer’s careful check. We have changed “trifluoroacetic acid” to “trifluoroacetic acid (TFA)” in the revised manuscript. Please check it out in the revised manuscript.
Point 4: line 364, "cuased"
Response 4: Line 364: Thanks for the reviewer’s careful check. We have changed “cuased” to “caused” in the revised manuscript. Please check it out in the revised manuscript.
Point 5: My previous concern on the digit of Mw was whether the Mw determined by HPGPC can be as accurate as 10 Da or not. please keep using 9023Da along with 7549 Da.
Response 5: Thanks for the reviewer’s suggestion. As the reviewer suggested, we have kept using 9023 Da along with 7549 Da in the revised manuscript. Please check it out in the revised manuscript.

This manuscript is a resubmission of an earlier submission. The following is a list of the peer review reports and author responses from that submission.